# Paediatric Spinal Deformity Surgery: Complications and Their Management

**DOI:** 10.3390/healthcare10122519

**Published:** 2022-12-13

**Authors:** Simon B. Roberts, Athanasios I. Tsirikos

**Affiliations:** 1Leeds Teaching Hospitals NHS Trust, Great George Street, Leeds LS1 3EX, UK; 2Scottish National Spine Deformity Centre, University of Edinburgh, Royal Hospital for Children and Young People-Edinburgh, 50 Little France Crescent, Edinburgh EH16 4TJ, UK

**Keywords:** complications, management, paediatric, spinal, surgery

## Abstract

Surgical correction of paediatric spinal deformity is associated with risks, adverse events, and complications that must be preoperatively discussed with patients and their families to inform treatment decisions, expectations, and long-term outcomes. The incidence of complications varies in relation to the underlying aetiology of spinal deformity and surgical procedure. Intraoperative complications include bleeding, neurological injury, and those related to positioning. Postoperative complications include persistent pain, surgical site infection, venous thromboembolism, pulmonary complications, superior mesenteric artery syndrome, and also pseudarthrosis and implant failure, proximal junctional kyphosis, crankshaft phenomenon, and adding-on deformity, which may necessitate revision surgery. Interventions included in enhanced recovery after surgery protocols may reduce the incidence of complications. Complications must be diagnosed, investigated and managed expeditiously to prevent further deterioration and to ensure optimal outcomes. This review summarises the complications associated with paediatric spinal deformity surgery and their management.

## 1. Introduction

The majority of children and adolescents with paediatric spinal deformity lead active lives and are healthy. The most common type of paediatric spinal deformity is scoliosis [1], which may be classified by aetiology as idiopathic, neuromuscular, congenital, syndromic, and scoliosis associated with cardiac disease or intraspinal anomalies. Idiopathic scoliosis can be further classified by age of onset into infantile (<3 years; IIS), juvenile (4 to 9 years; JIS), and adolescent (10 to 18 years; AIS), which is the most prevalent [2]. Scoliosis with onset before age of ten years may also be defined as early-onset scoliosis (EOS), including children with idiopathic, neuromuscular, congenital, thoracogenic, or syndromic aetiologies; late-onset scoliosis refers to patients with onset of idiopathic scoliosis after ten years of age [3]. Defining scoliosis as early- or late-onset may help focus treatment in relation to the patient’s physiological development [4]. Spinal deformities affecting primarily the sagittal plane include Scheuermann’s kyphosis and spondylolisthesis. Spinal deformity may progress during periods of skeletal growth and after skeletal maturity and be associated with back pain, respiratory dysfunction, patient concerns regarding self-image, and impaired quality of life [5,6]. Surgical management of spinal deformity in paediatric patients aims to safely correct spinal deformity and prevent future progression by achieving a solid spinal fusion.

Surgical correction of paediatric spinal deformity is associated with risks, adverse events, and complications. These must be anticipated, and their incidence, implications and management preoperatively conveyed clearly and concisely with patients and their families to inform treatment decisions, expectations, and long-term outcomes. The incidence of complications varies in relation to the underlying aetiology of spinal deformity and surgical procedure. The overall incidence of complications associated with surgical correction for patients with AIS has been reported between 5–23%, for neuromuscular scoliosis as 35%, and congenital scoliosis as 14% [7,8]. Complications may also be classified by their severity and timing in relation to surgery, whether occurring intraoperatively or postoperatively. This review will summarise the complications associated with paediatric spinal deformity surgery and their management.

## 2. Intraoperative Complications

### 2.1. Bleeding

Surgical correction of spinal deformities may be associated with significant blood loss and consequent risks including blood transfusion requirements, organ hypoperfusion, spinal cord hypoperfusion, and increased length of stay [9,10]. Allogeneic or autologous blood transfusion rates for paediatric patients undergoing surgical correction of spinal deformity have been reported at between 18.2–25.1% [11,12]. Intraoperative blood loss is greater in patients with neuromuscular scoliosis compared to those patients with AIS. Patients with Duchenne muscular dystrophy (DMD) are at particular risk of significant blood loss; patients with DMD are deficient of dystrophin in all muscle types and may have impaired vasoconstrictive responses [13]. Intraoperative blood loss is greater with increasing number of spinal levels instrumented, posterior compared to anterior spinal fusion, and for combined anterior and posterior surgery [14].

It is, therefore, critical to optimise haemoglobin level and coagulation profile preoperatively, with consultation with haematological specialists, and consideration of iron supplementation or recombinant erythropoietin [15,16]. During the positioning of patients prone for surgery, the abdomen must be kept free from direct pressure to avoid increased venous pressure in vertebral vessels and risk of increased intraoperative bleeding within the surgical field [17]. Local anaesthetic with epinephrine may be infiltrated prior to skin incision. Controlled hypotensive anaesthesia during surgical dissection decreases blood loss by 55%, transfusion requirements by 53% and mean operative duration by 81 min [18]. The use of topical haemostatic agents, intraoperative cell salvage, tranexamic acid (as bolus, infusion, and/or to soak surgical sponges), fibrinogen concentrate infusion, electrocautery throughout surgical dissection, bipolar tissue sealants, and ultrasonic bone scalpel for osteotomies may reduce intraoperative bleeding [19,20,21,22]. When used in conjunction with other blood conservation techniques, autologous blood transfusion wound drains used postoperatively lead to a reduced need for allogeneic blood transfusion in patients undergoing scoliosis surgery [23].

### 2.2. Neurological Injury

Neurological injury may range from transient peripheral nerve palsy to paralysis with complete spinal cord injuries. Neurological deficit may be due to vascular, metabolic, mechanical, or instrument-related complications [24]. Recent reports from the Scoliosis Research Society (SRS) Morbidity and Mortality Database identified the overall neurological deficit rate as 0.71–0.94% [25]. Patients undergoing surgery for congenital kyphosis, thoracic hyperkyphosis and needing corrective osteotomies are at increased risk of neurological complications [25,26].

Intraoperative neuromonitoring (IOM) provides routine contemporaneous recording of spinal cord function by utilising somatosensory (SSEPs) and transcranial motor evoked potentials (tcMEPs). Multimodal IOM is reported to provide 100% sensitivity in detection of spinal cord injury [27,28]. True events have identifiable precipitating factors, most of which can be reversed effectively [28]. Diagnostic criteria for IOM events that are true, transient, false, positive and negative, as well as decision algorithms, have been reported in response to MEP events during spinal surgery [27]. An intraoperative checklist has also been reported to optimise responses to IOM events when they occur [29]. Steps in these algorithms include stopping the operation and gaining control of the operating room and senior theatre personnel, optimising the mean arterial pressure (MAP)/haematocrit/blood pH and pCO_2_, seeking normothermia, discussing the potential need for the Stagnara wake-up test with anaesthetic staff, assessing anaesthetic agents/extent of neuromuscular blockade/paralysis, checking IOM electrodes and connections, determining timing and pattern of IOM signal changes, consultation with a colleague, and checking cervical and limb positions [30]. Surgical considerations include reviewing surgical steps prior to IOM signal changes and a consideration of reversing surgical manoeuvres (such as traction, distraction or corrective forces, removing rods or removing screws and probing screw tracks) to the time of last normal signals, assessing for spinal cord compression, and reviewing osteotomy and laminotomy sites [29]. If IOM signals recover, the surgical procedure may be completed if IOM signals remain stable. Consideration is needed to modify the surgical plan and to accept a more moderate correction with the prerequisite that IOM signals are stable [27]. The surgical procedure may need to be staged, and consideration given to administration of IV steroids [29]. If IOM signals do not recover, there is risk of permanent neurological deficit and consideration must be given to abandoning the procedure and removing all instrumentation [27]. The neurological status of the patient must be assessed on waking [27]. Intraoperative or peri-operative imaging (CT/MRI) should be considered to evaluate for neurological injury or compression, as well as the position of all instrumentation [30].

Neurological recovery has been reported in 87.7% of patients with neurological deficit following surgery for spinal deformity; 70.8% of these patients had complete recovery at long-term follow-up, with recovery occurring during the first one to two years postoperatively [24]. In patients with a preoperative neurological deficit, further insult to the spine can occur during surgery to correct deformity. The surgical strategy should, therefore, include moderate corrective manoeuvres to prevent progression of deformity and decompression to permit neurological recovery. Dural tear may occur during osteotomies, decompression or directly due to the placement of pedicle screw instrumentation; repair should include watertight closure with sutures or clips with or without supplemental fibrin glue and/or overlying patch sealant [31,32]. Traction may be utilised to perform gradual correction of spinal deformity, which may increase the tolerance of the spinal cord to subsequent corrective manoeuvres and definitive surgery [33,34].

The most common cause of neurological complications during surgery for paediatric spinal deformity is mechanical injury [35]. This includes cord compression by spinal instrumentation, haematoma, ligament, or bone. Complete neurological recovery has been reported following surgical decompression or removal of aberrant pedicle screws. Overcorrection, causing neurological compromise, can be reversed by loosening the spinal instrumentation [36]. The recovery of neurological impairment following an ischaemic insult to the spinal cord has been reported as less predictable, emphasising the importance of maintaining optimal MAP during surgery requiring osteotomies and extensive spinal instrumentation [24]. Following surgery with neurological deterioration, the optimisation of physiological parameters and active prevention of secondary complications may be appropriate for patients without spinal stenosis or cord compression [37]. Repeat neurological assessment and documentation during the postoperative recovery is fundamental. Delayed-onset postoperative neurological deficit has also been reported; CT and MR imaging is required to determine location of any spinal cord compression or malposition of spinal instrumentation to inform whether or not surgical intervention may be beneficial [38].

### 2.3. Positioning

Postoperative blindness or visual loss (POVL) are debilitating complications of surgery to correct paediatric spinal deformity. The incidence of POVL following surgery for paediatric spinal deformities has been reported as up to 0.03-0.16% [25,39]. Risk factors for POVL include inadequate patient positioning, increased blood loss, and long duration of surgery [39]. POVL can be avoided by the surgical team, anaesthetic team, and operating personnel ensuring that the patient’s eyes are free from any pressure. Paediatric patients undergoing spinal surgery are more likely than adults to develop non-ischaemic optic neuropathy and non-central retinal artery occlusion [40]. Spinal surgery of duration greater than 6.5 h, or blood loss greater than 44.7% of estimated blood volume, may place patients at high risk of POVL [41]. Other positioning-related complications include perioperative peripheral nerve injury (PPNI), which more frequently affects the brachial plexus, the ulnar, median or radial nerves, or the lateral femoral cutaneous nerve [42]. PPNI can be caused by direct pressure, stretch, and/or ischaemia of nerve fibres; these processes are often interdependent [43]. The brachial plexus is stressed most in positions of contralateral cervical spine flexion, lateral rotation of the shoulder, shoulder abduction and wrist extension [42]. Ulnar neuropathy may occur with the elbow kept flexed for prolonged length of time [42]. Elbow extension and wrist hyperextension may overstretch the median nerve; median neuropathy often leads to sustained dysfunction [42,44]. Radial nerve injury may occur from direct pressure on the arm, especially in the lateral position [42,44]. Careful preparation during patient positioning and protection of bony anatomical prominences with pads can protect against peripheral nerve palsies and brachial plexus injuries.

## 3. Postoperative Complications

### 3.1. Persistent Pain

Surgical correction of AIS is associated with an overall improvement in pain following surgery. However, a proportion of patients report moderate to severe pain persisting for more than one year following surgery [45]. Patients with persistent pain should be investigated for postoperative infection, inflammation, instrumentation failure or misplacement, pseudarthrosis, neurological injury, or progressive deformity. Prominent portions of instrumentation causing persistent pain may be identified by clinical examination in conjunction with radiographs. Definitive management of persistent pain relating to prominent implants will require the removal of instrumentation; this should be delayed until spinal fusion is established and confirmed. Effective multimodal analgesia in the postoperative period may prevent central sensitisation and development of persistent pain [46]. Persistent pain may lead to increased medication use, and school or work absences. Non-pharmacological interventions such as cognitive behavioural therapy (CBT), relaxation, and biofeedback have been helpful in treating paediatric patients with chronic pain [47].

### 3.2. Surgical Site Infection

Patients with neuromuscular scoliosis are at significantly greater risk of postoperative surgical site infection (SSI) and deep wound infection compared to patients with idiopathic scoliosis. Patients with neuromuscular scoliosis are at increased risk of SSIs due to loss of protective sensation in the lower back and extremities, reduced mobility, loss of bowel and/or bladder control, and compromised soft tissue due to previous surgeries [48,49].

Acute postoperative wound infection rates for patients with neuromuscular scoliosis have been reported to vary between 3.02–3.73% compared to 0.31–1.25% in patients with idiopathic scoliosis [25]. Overall wound infection rates for patients with neuromuscular scoliosis have been reported as 4.2–20%, and overall rates for deep infection following spinal deformity surgery have been reported at 2.82% [49,50]. Deep wound infections following corrective spinal surgery in patients with neuromuscular scoliosis are often caused by polymicrobial infection or gram negative bacteria with high virulence [51]. This may be due to more extensive posterior spinal wounds extending to the lumbosacral region in association with instrumentation to the pelvis, increased risk of wound contamination associated with bowel and bladder dysfunction, presence of VP shunt, and poor nutrition [52,53]. Patients with a diagnosis of myelodysplasia are at particularly high risk of postoperative SSIs, especially in the presence of a preoperative VP shunt, and due to fewer layers of the posterior soft tissue [52]. SSIs in patients with AIS are usually delayed in presentation (>6 months after corrective surgery) and are often caused by skin organisms with low virulence [54]. No difference has been demonstrated in rates of SSI with the use of different types of bone graft to promote spinal fusion [55,56].

Early SSIs (<6 months after primary corrective spinal surgery) may be managed by operative intervention with irrigation and debridement while retention of instrumentation is possible [7]. A discharging wound or haematoma are indications for surgical irrigation and debridement. Delay in surgical intervention may lead to clinical deterioration, sepsis or osteomyelitis [54,57]. Advice should be sought from infectious diseases or microbiology specialists to guide long-term antimicrobial therapy informed by the results from intraoperative sample cultures; antibiotics may need to be continued until spinal fusion is achieved [49]. The application of serial closed negative pressure dressing systems may promote development of granulation tissue over instrumentation and assist wound closure [58]. Delayed presentation (>6 months after primary corrective spinal surgery) of deep spinal infection in the presence of solid spinal fusion may be treated with the debridement, irrigation, removal of instrumentation and antimicrobial therapy [59]. Delayed presentation with uncertain fusion mass should be treated by surgical debridement, and if recurrent, implant removal [49]. Deep infection following corrective surgery for paediatric spinal deformity persists in half of patients with SSIs if instrumentation is not removed [60]. Progression of deformity may occur following the removal of instrumentation and is usually modest [59]. Patients may require bracing to minimise the loss of correction or re-instrumentation [61,62]. Further progression of deformity or pseudoarthrosis can be managed after clearance of infection.

Guidelines to prevent surgical site infection following spinal fusion surgery in high-risk paediatric patients have been developed using the Delphi process and an expert panel of paediatric spinal surgeons [63]. Recommendations from these guidelines are summarised in Table 1.

### 3.3. Venous Thromboembolism

There is limited literature published regarding the incidence of venous thromboembolism (VTE) or pulmonary embolism (PE) following paediatric spinal deformity surgery. The incidence of paediatric patients developing VTE following spinal fusion surgery for correction of AIS has been reported as 0.04% in a cohort of 21,955 paediatric patients that underwent spinal fusion surgery [64]. Risk factors for paediatric patients developing VTE following spinal fusion surgery include venous stasis peri-operatively, comorbidities, and coagulopathies. In paediatric patients undergoing spinal fusion surgery, advancing age, diagnosis of congenital or syndromic scoliosis or kyphoscoliosis have been associated with the development of VTE complications [64].

A concomitant hypercoagulable state has been demonstrated to be associated with a 14-fold increased incidence of VTE complications following spinal fusion surgery for AIS; 18.4% of patients with a pre-existing diagnosis of a hypercoagulable condition developed a VTE postoperatively [65]. In paediatric patients undergoing spinal fusion surgery, increasing age (15–19 years of age), obesity, and those requiring surgery involving 13 or more vertebral levels are also at increased risk of VTE complications. However, the overall incidence of VTE complications is very low at <1% in these patients [65]. As the overall incidence of VTE complications is very low, postoperative anticoagulation in paediatric patients undergoing spinal fusion surgery is not indicated [66]. For patients with a known hypercoagulable condition, assessment by a haematological specialist should be obtained preoperatively to inform need for anticoagulation and balance risks of increased bleeding and haematoma formation. Investigations for thrombophilia should be performed for paediatric patients with previous VTE, recurrent VTE, strong family history, VTE at uncommon vascular site, neonatal purpura fulminans, warfarin-induced skin necrosis, or recurrent pregnancy loss [65].

### 3.4. Pulmonary Complications

Scoliosis causes vertebral rotation and chest wall deformity by rotating the ribs. Pulmonary function tests (PFTs) may be performed to evaluate respiratory dysfunction, and to assess preoperative and postoperative lung function. The main thoracic curve severity, main thoracic rib prominence, apical vertebral translation and thoracic hypokyphosis are correlated with restrictive lung disease [67]. The three-dimensional thoracic deformity caused by severe scoliosis leads to lung function impairment due to secondary torsion of the diaphragm, impaired thoracic compliance, lung parenchymal compression, airway narrowing, and reduced lung volumes [68,69]. Paediatric patients with early onset scoliosis (EOS) are at particular risk of developing thoracic insufficiency syndrome due to the combination of spinal deformity, rib anomalies, diminished thoracic height, and restriction of spine and thoracic growth should early spinal fusion occur [70]. Up to 25% of paediatric patients with thoracic AIS demonstrated significant lung function impairment preoperatively [71]. Restrictive lung disease may occur with thoracic AIS greater than 70°, and significant predisposition to cardio-respiratory dysfunction with curves greater than 90° [72].

The incidence of pulmonary complications in the postoperative period following paediatric spinal deformity surgery has been reported as 0.6% for idiopathic scoliosis, 1.9% for neuromuscular scoliosis, and 1.1% for congenital scoliosis [73]; this included atelectasis, pneumonia, pulmonary oedema, pulmonary fat emboli, and respiratory failure. The incidence of pulmonary complications can be reduced by managing patients with neuromuscular scoliosis undergoing spinal deformity surgery using an accelerated discharge pathway [74].

Surgical correction of AIS results in stabilising or mildly improving pulmonary function [75]. Thoracic volume increases postoperatively, though this may also be attributed to continued physiological growth; increases in thoracic volume correspond with improvements in PFTs [76]. In AIS patients undergoing posterior spinal fusion, exercise testing and maximal oxygen uptake (VO2_max_) are maintained within normal parameters pre- and postoperatively. In AIS patients assessed at long-term follow-up, restrictive ventilatory defects were demonstrated in 27.7% of patients, which were associated with large rib hump and vertebral rotations [76]. Anterior instrumented surgery utilising intra-thoracic approaches are associated with decline in PFTs at 5 years postoperatively, whereas PFTs remain stable or improve following posterior surgery [77,78]. Pulmonary function has been shown to be maintained at mean 4.8 years postoperatively following thoracoplasty performed in conjunction with posterior spinal fusion [79]. Posterior instrumented spinal fusion, without chest wall disruption, to correct severe spinal deformity is associated with preserved lung function at 10 years postoperatively [80]. Abnormal chest wall motion associated with severe scoliosis can be improved by the use of non-invasive ventilation [81].

### 3.5. Superior Mesenteric Artery Syndrome (SMAS)

Superior mesenteric artery syndrome (SMAS) is defined as obstruction of the third part of the duodenum between the superior mesenteric artery (SMA) and the aorta. The duodenum is suspended by the ligament of Trietz. SMAS develops rarely following surgery to correct spinal deformity. Corrective spinal surgery lengthens the vertebral column and reduces the aortomesenteric angle which may lead to small bowel obstruction and SMAS. SMAS may also develop following disruption of the autonomic nerve supply to the small bowel during anterior approaches to the spine. Patients with increased risk of developing SMAS include those with BMI below 25th percentile for age, rigid thoracic scoliosis, laterally displaced lumbar curves, undergoing kyphoscoliosis correction, and the use of casts or halo-femoral traction [82,83].

SMAS may present with symptoms including bilious vomiting, abdominal pain, and early satiety. The main differential diagnosis to consider is paralytic ileus. SMAS usually develops within 1–2 weeks following corrective spinal surgery. SMAS may lead to delayed postoperative recovery, delayed nutritional recovery, wound healing problems, and prolonged in-patient stay [84].

CT imaging of the abdomen will demonstrate obstruction of the third part of the duodenum at the aorto-mesenteric angle and confirm the diagnosis of SMAS. Barium swallow radiography may also be used to demonstrate restriction at the third part of the duodenum or to confirm resolution of SMAS. SMAS can usually be treated successfully conservatively [83]. Conservative management of SMAS includes restriction of oral intake, nasogastric tube decompression of the stomach, maintaining fluid balance and correcting electrolyte abnormalities, antiemesis, administration of a prokinetic agent to promote bowel motility, dietician and gastrointestinal specialty reviews, nutritional support, positioning in left lateral decubitus or prone position, and daily weight measurements [83,85]. Insertion of a multilumen nasojejunal tube should be established to permit decompression and feeding; NJ feeding may be commenced and increased gradually to achieve weight gain. If enteral feeding is not possible, total parenteral nutrition should be commenced. When tube drainage decreases to <100 mL in 8 h, oral fluid administration may commence, progressing gradually to soft diet and more regular feeding [85,86]. The removal of spinal instrumentation is rarely required. If SMAS is due to lumbar hyperextension or casting, the removal of spinal instrumentation or cast may be required to improve the compression causing SMAS. Surgical intervention may be required for persistent symptoms, weight loss, bilious vomiting, or electrolyte abnormalities. Surgical intervention may require gastro-jejunostomy or duodenojejunostomy [83]. Patients that are underweight and that will require staged anterior and posterior spinal surgery for deformity may benefit from enteral feeding in the interval between the two procedures to prevent further weight loss that may contribute to risk of developing SMAS [85].

### 3.6. Early Recovery after Surgery Protocols

The use of enhanced recovery after surgery (ERAS) protocols incorporating multiple pre-, intra-, and postoperative interventions to improve patient care surrounding paediatric spinal deformity surgery is associated with a reduced length of stay, reduced rate of complications (reported reduction of up to 63%) and significantly less postoperative pain, and may lead to cost savings [87]. ERAS protocols have been reported for primary surgery to correct paediatric spinal deformity, but the interventions may also be applicable to revision procedures. A summary of interventions included in ERAS protocols is shown in Table 2.

### 3.7. Pseudarthrosis and Implant Failure

Pseudarthrosis may be suspected following corrective surgery for spinal deformity in the presence of persistent postoperative pain, loss of deformity correction, or instrumentation failure. Pseudarthrosis is the absence of solid bony fusion at least one year following correction of spinal deformity [91]. Risk factors for pseudarthrosis include an increased number of vertebral levels involved in the instrumented fusion, smoking, thoracolumbar kyphosis >20°, and fusion to the sacrum [92]. Paediatric patients most at risk of pseudarthrosis include patients with neuromuscular conditions, osteoporosis, nutritional deficiency or metabolic bone diseases [92,93]. Amongst patients with neuromuscular scoliosis, patients with myelodysplasia are at significantly increased risk of pseudarthrosis, associated with greater risk of surgical site infection, implant loosening, fusion to sacrum, and high correction loss [94,95]. Factors contributing to high risk of pseudarthrosis in patients with neuromuscular scoliosis include malabsorption syndrome, phosphate depletion, vitamin D abnormalities, and anaemia, all of which have detrimental effects on fusion rates [96]. Static radiographs allow for assessment of instrumentation integrity and for any gross loss of correction of spinal deformity. Dynamic radiographs may demonstrate translation or angulation at sites of pseudarthrosis. CT is most accurate for demonstrating successful fusion or sites of pseudarthrosis [97]. Skeletal scintigraphy (bone scanning) may also be helpful if CT is not conclusive; bone scanning may demonstrate increased radiotracer uptake at sites of pseudarthrosis and also demonstrate implant loosening—its specificity is limited as increased uptake may also be related to healing and remodelling and, therefore, bone scanning is more helpful when used more than one year following surgery [98].

Careful and thorough decortication across the levels of fusion and adequate use of bone graft will reduce the risk of pseudarthrosis [99]. Pseudarthrosis may also occur due to inadequate spinal balance, unfavourable forces at osteotomy sites or insufficient stability, especially at the lumbosacral junction and fixation to the pelvis. Delayed fusion or malunion may require bracing and reduced activities [100]. Established pseudarthrosis usually requires surgical intervention with consideration of the complete removal of interposed fibrous tissue, revision bone grafting with copious autologous bone supplemented by allograft bone or bone substitutes, optimisation of sagittal and coronal balance which may necessitate correctional osteotomies, and additional or segmental fixation with or without circumferential fusion [101]. Occult infection contributing to pseudarthrosis must always be considered. Intraoperative tissue samples should be routinely obtained at revision surgery and cultured to guide postoperative antimicrobial therapy [102].

Implant loosening is defined as a lucent rim of at least 2 mm surrounding the implant, usually at the tip of implanted pedicle screws. Osteolysis surrounding implants may also occur related to infection; osteolysis due to infection usually appears circumferentially around implants and in association with adjacent soft tissue oedema [103]. The instrumentation should always be reviewed for implant disengagement and rod or screw fractures [101].

### 3.8. Proximal Junctional Kyphosis

Proximal junctional kyphosis (PJK) is defined as a proximal junctional sagittal Cobb angle ≥10° and a proximal junctional sagittal Cobb angle at least 10° greater than the preoperative measurement, as measured between the caudal end plate of the upper instrumented vertebra (UIV) and the cephalad end plate of the two supra-adjacent vertebrae [104]. PJK has also been defined as any postoperative kyphosis increase ≥15° between the caudal endplate of the UIV and cephalad endplate of the single vertebra above the UIV [105]. In paediatric patients, PJK often occurs as a postoperative kyphotic change in the intervertebral disc cranial to the spinal instrumentation and fusion for spinal deformity. The incidence of PJK following corrective surgery for AIS has been reported as high as 46% [106]. Risk factors for developing PJK following spinal fusion for AIS include thoracoplasty, preoperative thoracic hyperkyphosis, the use of hybrid instrumentation, the use of pedicle screws at the UIV, autogenous bone graft and distal fusion below L2, the use of combined anterior-posterior instrumentation compared to posterior instrumentation only, and instrumentation to the sacrum [107,108,109]. The incidence of PJK following surgical correction of Scheuermann kyphosis has been reported between 3–34.4% [110,111]. The incidence of PJK following surgical correction of spinal deformity in patients with neuromuscular scoliosis has been reported at 27% [112]. Patients with neuromuscular scoliosis at increased risk of developing PJK include those with greater magnitude of sagittal profile, lumbar lordosis, BMI, shoulder imbalance, loss of major coronal curve correction, and the use of halo gravity traction perioperatively [112]. Development of radiographic PJK correlates poorly with clinical outcomes, and few paediatric patients require revision surgery to extend the fusion and achieve global sagittal balance.

Surgical management of clinically significant PJK requires the extension of the posterior spinal fusion to include the affected segments; decompression may be required for advanced PJK, and anterior approaches may be considered for spondylolisthesis or kyphosis requiring anterior structural support [101]. Increased junctional angle may lead to a higher revision rate due to implant failure and pseudarthrosis [113]. PJK may be minimised by preserving the interspinous ligament, facet capsules and fascia above the upper instrumented level. For patients with AIS, PJK may be minimised by restoring thoracic kyphosis, and the use of proximal hooks rather than pedicle screws [114]. For patients with Scheuermann kyphosis, PJK may be minimised by preventing overcorrection of thoracic hyperkyphosis, especially if the patient has high pelvic incidence, and including the proximal end vertebra within the fusion construct [115]. Further strategies to prevent the development of PJK may include the use of transition rods to reduce stress proximal to the fused spine, and minimising cantilever forces at the proximal extent of the instrumented construct [114].

### 3.9. Crankshaft Phenomenon

The crankshaft phenomenon is a progressive rotational and angular spinal deformity that can occur after posterior spinal surgery in skeletally immature patients [116]. This can result in progressive spinal imbalance following an initial satisfactory surgical correction of spinal deformity and develops due to disproportionate anterior vertebral growth in the spinal segments included in the posterior fusion. Patients most at risk include those undergoing posterior spinal fusion prior to their adolescent growth spurt, with open triradiate cartilage, and at Risser stage 0–1 [117,118]. Crankshaft deformity may develop in patients with scoliosis of all causes. Surgical management may be required for significant spinal imbalance leading to functional limitation and poor cosmetic appearance.

Anterior apical release and fusion was originally reported to significantly limit the risk of developing a crankshaft deformity [119]. The effectiveness of segmental fixation with pedicle screws systems to prevent crankshaft deformity has also been reported, with favourable outcomes for spinal constructs utilising pedicle screws rather than hooks for posterior segmental instrumentation to manage patients with AIS and Risser grade 0 [120,121]. Rigid instrumentation with pedicle screw constructs is effective in preventing crankshaft deformity in surgical correction of AIS and JIS [120,121,122]. In patients with cerebral palsy and open triradiate cartilage, the Unit Rod system with sublaminar wires has been reported to provide sufficient rigidity to prevent the development of crankshaft deformity [123]. Growth-friendly surgical interventions for the management of early-onset scoliosis may increase the risk of posterior autofusion and risk of crankshaft phenomenon; controlling apical scoliotic deformity and permitting spinal growth may limit the risk of developing crankshaft deformity [124]. In skeletally immature patients with crankshaft deformity, anterior spinal fusion may be considered. Chest wall deformities may require thoracoplasty. Severe rotational crankshaft deformity may require implant removal, testing of the fusion mass, multicolumn spinal osteotomies, and revision instrumented fusion [125].

### 3.10. Adding-On Deformity

Adding-on deformity is the progression of spinal deformity in the adjacent un-instrumented and unfused spinal segments following spinal fusion surgery. It may occur due to inadequate selection of end vertebrae in the surgical treatment of AIS, especially when managing major thoracic curves with compensatory lumbar curves by selective thoracic fusion [101]. Significant progression of spinal deformity will require correction with extension of instrumented fusion and should be performed without delay when detected.

### 3.11. Revision Surgery

The revision surgery to correct paediatric spinal deformity will be determined by the nature of the initial procedure and of any complications, as well as the patient’s underlying medical conditions and clinical presentation. The rate of revision surgery in a national spinal deformity service has been reported as 2.9% for removal or exchange of instrumentation, 1.5% for nonunion, 1.1% for infection, and 0.8% for adding-on deformity requiring extension of fusion [1]. Traction in the form of halo-gravity or skull-femoral traction may be used preoperatively, intraoperatively or between staged procedures. Traction may be helpful for patients who require gradual deformity correction prior to revision surgery [126]. Corrective posterior osteotomies include the Smith-Peterson osteotomy, pedicle subtraction osteotomy, and posterior vertebral column resection [101]. A Smith-Peterson osteotomy (SPO) involves the fracture and resection of the posterior fusion mass at the level of the lamina and facets; this can be performed at multiple levels and may result in 10–15° lordotic correction at each level. Multiple SPOs are useful to correct gradual deformity over several levels [127]. Pedicle subtraction osteotomy (PSO) is a three-column posterior wedge osteotomy that permits up to 30° of correction, and can be customised to achieve sagittal, coronal or multi-planar correction [101,128]. Vertebral column resection (VCR) can correct severe spinal deformity. VCR permits coronal, sagittal and axial plane correction of deformity and anterior decompression. VCR involves complete vertebrectomy, can be performed by a posterior costo-transversectomy approach, and requires the stabilisation of the vertebral column during and after resection along with anterior column reconstruction [129]. PSO is particularly effective for correcting kyphoscoliosis or lumbar lordosis, and VCR is particularly effective for severe focal spinal deformity requiring shortening or rotational correction [130].

### 3.12. Navigation and Robotic-Assisted Surgery

Computer-assisted navigation (CAN), and robotic-assisted (RA) techniques have been reported to improve the accuracy of pedicle screw placement compared to freehand techniques but are associated with no improvement in complication rates or patient outcomes scores [131,132,133,134,135,136]. Operative time and inpatient stay are longer, and patient exposure to radiation greater, with CAN and RA surgery compared to freehand techniques [132,137,138]. Freehand positioning of pedicle screws is safe and effective for correction of paediatric spinal deformity and associated with a significantly reduced radiation exposure to patients [139]. Further research and innovation will help to determine the optimal role of CAN and RA techniques in improving outcomes in the surgical correction of paediatric spinal deformity.

## 4. Conclusions

Surgery to correct paediatric spinal deformity is associated with rare but severe complications. Surgeons and healthcare professionals caring for paediatric patients undergoing surgery for spinal deformity must be familiar with the potential complications and their treatment. Patients and their families must also be informed regarding risk of complications and their management when consenting to surgery. Optimising medical comorbidities and education of patients and their families preoperatively, as well as meticulous surgical planning with particular attention to regional and global spinal alignment, can minimise complication rates. Complications must be diagnosed, investigated and managed expeditiously to prevent further deterioration and to ensure optimal outcomes.

## Figures and Tables

**Table 1 healthcare-10-02519-t001:** Recommendations to reduce surgical site infection following spinal fusion surgery in paediatric patients (adapted from reference [63]); IV—intravenous, UV—ultraviolet.

Chlorhexidine skin wash the night before surgery Preoperative urine cultures Preoperative patient education sheet Preoperative nutritional assessment If removing hair, clipping is preferred to shaving Peri-operative IV cefazolin Peri-operative IV prophylaxis for gram-negative bacilli Adherence to peri-operative antimicrobial regimens Operating room access should be limited during scoliosis surgery UV lights need not be used in the operating room Intraoperative wound irrigation Vancomycin powder should be used in bone graft and/or surgical site Impervious dressings are preferred postoperatively Dressing changes should be minimised before discharge

**Table 2 healthcare-10-02519-t002:** Interventions for ERAS protocols in paediatric spinal deformity surgery (adapted from references [87,88]); TIVA: total intravenous anaesthesia, MAP: mean arterial pressure, PCA: patient-controlled anaesthesia. * Power tools include power-assisted pedicle tract preparation, pedicle tapping, and pedicle screw insertion [89,90].

Preoperative	Intraoperative	Postoperative
Patient, family and carer education and expectation management Oral haematinics and multivitamins Chlorhexidine wash night before surgery Spinal physiotherapy Multimodal analgesia	Anaesthesia protocol may include: ○ Isovolumic haemodilution ○ TIVA ○ MAP maintained at 55–80 mm Hg Prophylactic antibiotics IV tranexamic acid Cell salvage use Refine implant density Consider the use of power tools for pedicle screw insertion * Dual consultant surgeon operating Limit fluoroscopy use Regional and local anaesthesia (intrathecal morphine, subcutaneous bupivacaine infiltration)	Day 1 postoperatively or as early as tolerated: PCA discontinuation and transition to multimodal oral analgesia Mobilisation and physiotherapy Removal of urinary catheter Removal of wound drain Resumption of oral diet Continued antiemesis Incentive spirometry Bowel care regimen

## Data Availability

Not applicable.

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
