# Peer review of "Paediatric Spinal Deformity Surgery: Complications and Their Management"

_healthcare, 2022, doi:10.3390/healthcare10122519_

Round 1

Reviewer 1 Report

lines 32, 65, 80, 92-111, 118123, 138, 152-160, 177, 184, 193, 195, 200, 229, 236, 238, 244, 260 277, 298, 317, 342, 352-363, 368, 394, 402-410, 429-434, 447, 449, 456 - reference should be added

lines 30-32 - classification to EOS (early onset scoliosis) and late-onset scoliosis is more commonly used nowadays. It's worth to be mentioned.

Author Response

REVIEWER 1 – COMMENTS

Comment 1: lines 32, 65, 80, 92-111, 118-123, 138, 152-160, 177, 184, 193, 195, 200, 229, 236, 238, 244, 260 277, 298, 317, 342, 352-363, 368, 394, 402-410, 429-434, 447, 449, 456 - reference should be added

Authors’ Response: Thank you for recommending references for these sections of the manuscript. The following references have been added for these sections of the manuscript (line numbers in brackets are the updated line numbers in the revised manuscript):

Line 32 (32):

  1. Yaman O, Dalbayrak S. Idiopathic scoliosis. Turk Neurosurg. 2014;24(5):646-57

Line 65 (70):

  1. Samdani AF, Torre-Healy A, Asghar J, Herlich AM, Betz RR. Strategies to reduce blood loss during posterior spinal fusion for neuromuscular scoliosis: a review of current techniques and experience with a unique bipolar electrocautery device. Surg Technol Int. 2008;17:243-8.

  1. Switzer T, Naraine N, Chamlati R, Lau W, McVey MJ, Zaarour C, et al. Association between preoperative hemoglobin levels after iron supplementation and perioperative blood transfusion requirements in children undergoing scoliosis surgery. Paediatr Anaesth. 2020;30(10):1077-82.

Line 80 (86):

  1. Li J, Hu Z, Qian Z, Tang Z, Qiu Y, Zhu Z, et al. The prognosis and recovery of major postoperative neurological deficits after corrective surgery for scoliosis : an analysis of 65 cases at a single institution. The bone & joint journal. 2022;104-B(1):103-11.

Line 92-111 (98-118):

  1. Tsirikos AI, Duckworth AD, Henderson LE, Michaelson C. Multimodal Intraoperative Spinal Cord Monitoring during Spinal Deformity Surgery: Efficacy, Diagnostic Characteristics, and Algorithm Development. Med Princ Pract. 2020;29(1):6-17.

  1. Vitale MG, Skaggs DL, Pace GI, Wright ML, Matsumoto H, Anderson RC, et al. Best Practices in Intraoperative Neuromonitoring in Spine Deformity Surgery: Development of an Intraoperative Checklist to Optimize Response. Spine Deform. 2014;2(5):333-9.

  1. Zuccaro M, Zuccaro J, Samdani AF, Pahys JM, Hwang SW. Intraoperative neuromonitoring alerts in a pediatric deformity center. Neurosurg Focus. 2017;43(4):E8.

Line 118-123 (125-130):

  1. West JL, Arnel M, Palma AE, Frino J, Powers AK, Couture DE. Incidental durotomy in the pediatric spine population. J Neurosurg Pediatr. 2018;22(5):591-4.

  1. Esposito F, Angileri FF, Kruse P, Cavallo LM, Solari D, Esposito V, et al. Fibrin Sealants in Dura Sealing: A Systematic Literature Review. PLoS One. 2016;11(4):e0151533.

  1. Sucato DJ. Management of severe spinal deformity: scoliosis and kyphosis. Spine. 2010;35(25):2186-92.

  1. Mehrpour S, Sorbi R, Rezaei R, Mazda K. Posterior-only surgery with preoperative skeletal traction for management of severe scoliosis. Archives of orthopaedic and trauma surgery. 2017;137(4):457-63.

Line 138 (146):

  1. Pizones J, Sponseller PD, Izquierdo E, Sanz E, Sanchez-Mariscal F, Alvarez P, et al. Delayed Tetraplegia After Thoracolumbar Scoliosis Surgery in Stuve-Wiedemann Syndrome. Spine Deform. 2013;1(1):72-8.

Line 148 (156):

  1. Patil CG, Lad EM, Lad SP, Ho C, Boakye M. Visual loss after spine surgery: a population-based study. Spine. 2008;33(13):1491-6.

Lines 152-160 (160-169):

  1. Kamel I, Barnette R. Positioning patients for spine surgery: Avoiding uncommon position-related complications. World J Orthop. 2014;5(4):425-43.

  1. Winfree CJ, Kline DG. Intraoperative positioning nerve injuries. Surgical neurology. 2005;63(1):5-18

  1. American Society of Anesthesiologists Task Force on Prevention of Perioperative Peripheral N. Practice advisory for the prevention of perioperative peripheral neuropathies: an updated report by the American Society of Anesthesiologists Task Force on prevention of perioperative peripheral neuropathies. Anesthesiology. 2011;114(4):741-54.

Line 177 (185):

  1. Ehde DM, Dillworth TM, Turner JA. Cognitive-behavioral therapy for individuals with chronic pain: efficacy, innovations, and directions for research. Am Psychol. 2014;69(2):153-66.

Line 184 (192-193):

  1. Janjua MB, Toll B, Ghandi S, Sebert ME, Swift DM, Pahys JM, et al. Risk Factors for Wound Infections after Deformity Correction Surgery in Neuromuscular Scoliosis. Pediatric neurosurgery. 2019;54(2):108-15.

  1. Bachy M, Bouyer B, Vialle R. Infections after spinal correction and fusion for spinal deformities in childhood and adolescence. International orthopaedics. 2012;36(2):465-9.

Line 193 (208):

  1. Clark CE, Shufflebarger HL. Late-developing infection in instrumented idiopathic scoliosis. Spine. 1999;24(18):1909-12.

Line 195 (212):

  1. Murphy RF, Mooney JF, 3rd. Complications following spine fusion for adolescent idiopathic scoliosis. Curr Rev Musculoskelet Med. 2016;9(4):462-9.

Line 200 (217):

  1. Bachy M, Bouyer B, Vialle R. Infections after spinal correction and fusion for spinal deformities in childhood and adolescence. International orthopaedics. 2012;36(2):465-9.

Line 229 (245):

  1. Jain A, Karas DJ, Skolasky RL, Sponseller PD. Thromboembolic complications in children after spinal fusion surgery. Spine. 2014;39(16):1325-9.

Line 236 (252):

  1. Rudic TN, Moran TE, Kamalapathy PN, Werner BC, Bachmann KR. Venous Thromboembolic Events are Exceedingly Rare in Spinal Fusion for Adolescent Idiopathic Scoliosis. Clin Spine Surg. 2022.

Line 238 (254):

  1. Erkilinc M, Clarke A, Poe-Kochert C, Thompson GH, Hardesty CK, O'Malley N, et al. Is There Value in Venous Thromboembolism Chemoprophylaxis After Pediatric Scoliosis Surgery? A 28-Year Single Center Study. Journal of pediatric orthopedics. 2021;41(3):138-42.

Line 244 (260):

  1. Rudic TN, Moran TE, Kamalapathy PN, Werner BC, Bachmann KR. Venous Thromboembolic Events are Exceedingly Rare in Spinal Fusion for Adolescent Idiopathic Scoliosis. Clin Spine Surg. 2022.

Line 260 (72):

  1. Sheehan DD, Grayhack J. Pulmonary Implications of Pediatric Spinal Deformities. Pediatric clinics of North America. 2021;68(1):239-59.

This sentence has also been amended to read ‘…and significant predisposition to cardio-respiratory dysfunction occurs with curves greater than 90o.

Line 277 (293):

Previous Lines 277-280 have been amended to read as follows, with two references added:

Three-dimensional correction of the spine and severe thoracic deformity is critical to minimise restrictive lung disease in decades following spinal deformity surgery. Anterior instrumented surgery utilising intra-thoracic approaches are associated with decline in PFTs at 5 years postoperatively, whereas PFTs remain stable or improve following posterior surgery (64, 70). Pulmonary function is has been shown to be maintained at mean 4.8 years postoperatively following thoracoplasty performed in conjunction with posterior spinal fusion (71). Posterior instrumented spinal fusion, without chest wall disruption, to correct severe spinal deformity is associated with preserved lung function at 10 years postoperatively (72).’

  1. Kim YJ, Lenke LG, Bridwell KH, Cheh G, Whorton J, Sides B. Prospective pulmonary function comparison following posterior segmental spinal instrumentation and fusion of adolescent idiopathic scoliosis: is there a relationship between major thoracic curve correction and pulmonary function test improvement? Spine. 2007;32(24):2685-93.

  1. Gitelman Y, Lenke LG, Bridwell KH, Auerbach JD, Sides BA. Pulmonary function in adolescent idiopathic scoliosis relative to the surgical procedure: a 10-year follow-up analysis. Spine. 2011;36(20):1665-72.

Line 298 (314):

  1. Voleti SPR, Sridhar J. Superior Mesenteric Artery Syndrome after Kyphosis Correction - A Case Report. J Orthop Case Rep. 2017;7(5):67-70.

Line 317 (333):

  1. Araujo AO, Oliveira RG, Arraes AJC, Mamare EM, Rocha ID, Gomes CR. Superior Mesenteric Artery Syndrome - An Uncommon Complication After Surgical Corrections of Spinal Deformities. Rev Bras Ortop (Sao Paulo). 2021;56(4):523-7.

Line 342 (360):

  1. How NE, Street JT, Dvorak MF, Fisher CG, Kwon BK, Paquette S, et al. Pseudarthrosis in adult and pediatric spinal deformity surgery: a systematic review of the literature and meta-analysis of incidence, characteristics, and risk factors. Neurosurg Rev. 2019;42(2):319-36.

  1. Rezende R, Cardoso IM, Leonel RB, Perim LG, Oliveira TG, Jacob Junior C, et al. Bone mineral density evaluation among patients with neuromuscular scoliosis secondary to cerebral palsy. Rev Bras Ortop. 2015;50(1):68-71.

Line 352-363 (375-386):

  1. Isik M, Ozdemir HM, Sakaogullari A, Cengiz B, Aydogan NH. The efficacy of in situ local autograft in adolescent idiopathic scoliosis surgery: a comparison of three different grafting methods. Turk J Med Sci. 2017;47(6):1728-35.

  1. McMaster MJ, James JI. Pseudoarthrosis after spinal fusion for scoliosis. The Journal of bone and joint surgery British volume. 1976;58(3):305-12.

  1. Kim HJ, Cunningham ME, Boachie-Adjei O. Revision spine surgery to manage pediatric deformity. The Journal of the American Academy of Orthopaedic Surgeons. 2010;18(12):739-48.

  1. Burkhard MD, Loretz R, Uckay I, Bauer DE, Betz M, Farshad M. Occult infection in pseudarthrosis revision after spinal fusion. The spine journal : official journal of the North American Spine Society. 2021;21(3):370-6.

Line 368 (392):

  1. Kim HJ, Cunningham ME, Boachie-Adjei O. Revision spine surgery to manage pediatric deformity. The Journal of the American Academy of Orthopaedic Surgeons. 2010;18(12):739-48.

Line 394 (423):

  1. Cho SK, Kim YJ, Lenke LG. Proximal Junctional Kyphosis Following Spinal Deformity Surgery in the Pediatric Patient. The Journal of the American Academy of Orthopaedic Surgeons. 2015;23(7):408-14.

Line 402-410 (432-440):

  1. Dubousset J, Herring JA, Shufflebarger H. The crankshaft phenomenon. Journal of pediatric orthopedics. 1989;9(5):541-50.

  1. Sanders JO, Little DG, Richards BS. Prediction of the crankshaft phenomenon by peak height velocity. Spine. 1997;22(12):1352-6; discussion 6-7.

  1. Roberto RF, Lonstein JE, Winter RB, Denis F. Curve progression in Risser stage 0 or 1 patients after posterior spinal fusion for idiopathic scoliosis. Journal of pediatric orthopedics. 1997;17(6):718-25.

Line 429-434 (459-464):

  1. Kim HJ, Cunningham ME, Boachie-Adjei O. Revision spine surgery to manage pediatric deformity. The Journal of the American Academy of Orthopaedic Surgeons. 2010;18(12):739-48.

Line 447 (477):

  1. Xia L, Li P, Wang D, Bao D, Xu J. Spinal osteotomy techniques in management of severe pediatric spinal deformity and analysis of postoperative complications. Spine. 2015;40(5):E286-92.

Line 449 (479):

  1. Kim HJ, Cunningham ME, Boachie-Adjei O. Revision spine surgery to manage pediatric deformity. The Journal of the American Academy of Orthopaedic Surgeons. 2010;18(12):739-48.

Diab MG, Franzone JM, Vitale MG. The role of posterior spinal osteotomies in pediatric spinal deformity surgery: indications and operative technique. Journal of pediatric orthopedics. 2011;31(1 Suppl):S88-98.

Line 456 (486):

  1. Ould-Slimane M, Hossein Nabian M, Simon AL, Happiette A, Julien-Marsollier F, Ilharreborde B. Posterior vertebral column resection for pediatric rigid spinal deformity. Orthopaedics & traumatology, surgery & research : OTSR. 2022;108(6):102797.

Comment 2: lines 30-32 - classification to EOS (early onset scoliosis) and late-onset scoliosis is more commonly used nowadays. It's worth to be mentioned.

Authors’ Response: Thank you for this recommendation. We have amended the article as follows to include discussion of definition of scoliosis as early-onset scoliosis (EOS) and late-onset scoliosis:

Page 1, paragraph 1, lines 32-37:

‘Scoliosis with onset before age of ten years may also be defined as early-onset scoliosis (EOS), including children with idiopathic, neuromuscular, congenital, thoracogenic, or syndromic aetiologies; late-onset scoliosis refers to patients with onset of idiopathic scoliosis after ten years of age (3). Defining scoliosis as early or late-onset may help focus treatment in relation to the patient’s physiological development (4).’

References added:

  1. Skaggs DL, Guillaume, T., El-Hawary, R., Emans, J., Mendelow, M., Smith, J. Early Onset Scoliosis Consensus Statement. Spine Deformity. 2015;3:107.
  2. Fletcher ND, Bruce RW. Early onset scoliosis: current concepts and controversies. Curr Rev Musculoskelet Med. 2012;5(2):102-10.

Reviewer 2 Report

Dear authors,

thank you for the opportunity to read this extensive review. Following points need some clarification in my opinion.

Lines 71-76: Various measures are mentioned about reducing intraoperative bleeding. One option not described ist the substitution of fibrinogen through infusion.

3.2 Surgical site infection. Was the role of bone substitute substances in surgical infection also researced?

Lines 189-193: Pathogens for postoperative infection in neuromuscular scoliosis and IAS are clearly different, however to a non-specialist it is not obvious why and the formulation awakes the false impression of a direct relation between the type of scoliosis and pathogen. A description of the reasons for this difference at this point could be usefull (eg wound extending to lumbosacral region in neuromuscular scoliosis and therefore higher proximity to anal region, often in combination with incontinence and other relevant reasons).

Line 341: Neuromuscular conditions are mentioned as a risk for pseudarthrosis. Which conditions are meant exactly? For example wheelchair bound patients and patients with reduced mobility can only exert little stress on the implants to cause a pseudarthrosis.

Table 2, Page 8: "Consider power tools for pedicle screw insertion". What is meant by "power tools"? Battery driven drills?

3.8 Proximal junctional kyphosis: Risk factors for neuromascular scoliosis are not refered to. Could you find any relevant reference to this aspect? For example limited or no head/neck muscle control can prove a major problem.

A last important point, that maybe should be mentioned, is the role of navigational surgery as a measure against complications in spinal deformity correction through safer and sturdier placement of pedicle screws.

Author Response

REVIEWER 2 - COMMENTS

Comment 1: Lines 71-76: Various measures are mentioned about reducing intraoperative bleeding. One option not described is the substitution of fibrinogen through infusion.

Authors’ Response: Thank you for this recommendation. We have amended the text (highlighted by italics) to include use of fibrinogen infusion in management of perioperative blood loss, and added the following reference (lines 76-79):

‘The use of topical haemostatic agents, intra-operative cell salvage, tranexamic acid (as bolus, infusion, and/or to soak surgical sponges), fibrinogen concentrate infusion, electrocautery throughout surgical dissection, bipolar tissue sealants, and ultrasonic bone scalpel for osteotomies may reduce intra-operative bleeding (19-22)’.

References added:

  1. Kozek-Langenecker S, Sorensen B, Hess JR, Spahn DR. Clinical effectiveness of fresh frozen plasma compared with fibrinogen concentrate: a systematic review. Crit Care. 2011;15(5):R239.

Comment 2: 3.2 Surgical site infection. Was the role of bone substitute substances in surgical infection also researched?

Authors’ Response: Thank you for this recommendation. We have added a comment with references regarding bone graft and surgical site infection, as follows (lines 208-209):

‘No difference has been demonstrated in rates of SSI with use of different types of bone graft to promote spinal fusion (55, 56).’

References added:

  1. Theologis AA, Tabaraee E, Lin T, Lubicky J, Diab M, Spinal Deformity Study G. Type of bone graft or substitute does not affect outcome of spine fusion with instrumentation for adolescent idiopathic scoliosis. Spine. 2015;40(17):1345-51.
  2. Kirzner N, Hilliard L, Martin C, Quan G, Liew S, Humadi A. Bone graft in posterior spine fusion for adolescent idiopathic scoliosis: a meta-analysis. ANZ J Surg. 2018;88(12):1247-52.

Comment 3: Lines 189-193: Pathogens for postoperative infection in neuromuscular scoliosis and IAS are clearly different, however to a non-specialist it is not obvious why and the formulation awakes the false impression of a direct relation between the type of scoliosis and pathogen. A description of the reasons for this difference at this point could be usefull (eg wound extending to lumbosacral region in neuromuscular scoliosis and therefore higher proximity to anal region, often in combination with incontinence and other relevant reasons).

Authors’ Response: Thank you for this comment. We have added the following comments (highlighted by italics) and references to provide greater explanation of reasons for difference in surgical site infection pathogens following surgery for patients with neuromuscular scoliosis and idiopathic scoliosis, and which patients with neuromuscular scoliosis are at particularly increased risk (lines 198-206):

‘Deep wound infections following corrective spinal surgery in patients with neuromuscular scoliosis are often caused by polymicrobial infection or gram negative bacteria with high virulence (51). This may be due to more extensive posterior spinal wounds extending to the lumbosacral region in association with instrumentation to the pelvis, increased risk of wound contamination associated with bowel and bladder dysfunction, presence of VP shunt, and poor nutrition (52, 53). Patients with a diagnosis of myelodysplasia are at particularly high risk for postoperative SSIs, especially in the presence of a preoperative VP shunt, and due to fewer layers of the posterior soft tissue (52).’

References added:

  1. Master DL, Poe-Kochert C, Son-Hing J, Armstrong DG, Thompson GH. Wound infections after surgery for neuromuscular scoliosis: risk factors and treatment outcomes. Spine. 2011;36(3):E179-85.
  2. Mackenzie WG, Matsumoto H, Williams BA, Corona J, Lee C, Cody SR, et al. Surgical site infection following spinal instrumentation for scoliosis: a multicenter analysis of rates, risk factors, and pathogens. The Journal of bone and joint surgery American volume. 2013;95(9):800-6, S1-2.

Comment 4: Line 341: Neuromuscular conditions are mentioned as a risk for pseudarthrosis. Which conditions are meant exactly? For example wheelchair bound patients and patients with reduced mobility can only exert little stress on the implants to cause a pseudarthrosis.

Authors’ Response: Thank you for this recommendation. We have added the following comments (highlighted by italics) and reference to provide greater explanation as to which patients with neuromuscular conditions are at increased risk of pseudarthrosis (lines 360-366):

‘Amongst patients with neuromuscular scoliosis, patients with myelodysplasia are at significantly increased risk for pseudarthrosis, associated with greater risk of surgical site infection, implant loosening, fusion to sacrum, and high correction loss (93, 94). Factors contributing to high risk of pseudarthrosis in patients with neuromuscular scoliosis include malabsorption syndrome, phosphate depletion, vitamin D abnormalities, and anaemia, all of which have detrimental effects on fusion rates (95).’

References added:

  1. Banit DM, Iwinski HJ, Jr., Talwalkar V, Johnson M. Posterior spinal fusion in paralytic scoliosis and myelomeningocele. Journal of pediatric orthopedics. 2001;21(1):117-25.
  2. Geiger F, Parsch D, Carstens C. Complications of scoliosis surgery in children with myelomeningocele. Eur Spine J. 1999;8(1):22-6.
  3. Steinmann JC, Herkowitz HN. Pseudarthrosis of the spine. Clinical orthopaedics and related research. 1992(284):80-90.

Comment 5: Table 2, Page 8: "Consider power tools for pedicle screw insertion". What is meant by "power tools"? Battery driven drills?

Authors’ Response: Thank you for this comment. We have now provided the following explanation (and references) in the Table 2 legend, using a star within the table to indicate additional information (lines 349-350):

‘*Power tools include power-assisted pedicle tract preparation, pedicle tapping, and pedicle screw insertion (88, 89).’

References added:

  1. Faldini C, Viroli G, Fiore M, Barile F, Manzetti M, Di Martino A, et al. Power-assisted pedicle screws placement: Is it as safe and as effective as manual technique? Narrative review of the literature and our technique. Musculoskelet Surg. 2021;105(2):117-23.
  2. Skaggs DL, Compton E, Vitale MG, Garg S, Stone J, Fletcher ND, et al. Power versus manual pedicle tract preparation: a multi-center study of early adopters. Spine Deform. 2021;9(5):1395-402.

Comment 6: 3.8 Proximal junctional kyphosis: Risk factors for neuromuscular scoliosis are not referred to. Could you find any relevant reference to this aspect? For example limited or no head/neck muscle control can prove a major problem.

Authors’ Response: Thank you for this recommendation. We have added 2 sentences (and references) describing the incidence of proximal junctional kyphosis in patients with neuromuscular scoliosis, and the risk factors for developing PJK in these patients, as follows (lines 408-413):

‘The incidence of PJK following surgical correction of spinal deformity in patients with neuromuscular scoliosis has been reported at 27% (111). Patients with neuromuscular scoliosis at increased risk of developing PJK include those with greater magnitude of sagittal profile, lumbar lordosis, BMI, shoulder imbalance, loss of major coronal curve correction, and use of halo gravity traction perioperatively (111).’

References added:

  1. Toll BJ, Gandhi SV, Amanullah A, Samdani AF, Janjua MB, Kong Q, et al. Risk Factors for Proximal Junctional Kyphosis Following Surgical Deformity Correction in Pediatric Neuromuscular Scoliosis. Spine. 2021;46(3):169-74.

Comment 7: A last important point, that maybe should be mentioned, is the role of navigational surgery as a measure against complications in spinal deformity correction through safer and sturdier placement of pedicle screws.

Authors’ Response: Thank you for the recommendation. We have now added a new section at the end of the manuscript ‘3.12 Navigation and robotic-assisted surgery’ to discuss the role of navigation and robotic-assisted techniques in the surgical correction of paediatric spinal deformity. The following text and references have been added (lines 492-502):

‘3.12 Navigation and robotic-assisted surgery

Computer-assisted navigation (CAN), and robotic-assisted (RA) techniques have been reported to improve the accuracy of pedicle screw placement compared to freehand techniques, but are associated with no improvement in complication rates or patient outcomes scores (130-135). Operative time and inpatient stay are longer, and patient exposure to radiation greater, with CAN and RA surgery compared to freehand techniques (131, 136, 137). Freehand positioning of pedicle screws is safe and effective for correction of paediatric spinal deformity, and associated with a significantly reduced radiation exposure to patients (138). Further research and innovation will help to determine the optimal role of CAN and RA techniques in improving outcomes in the surgical correction of paediatric spinal deformity.’  

References added:

  1. Campbell DH, McDonald D, Araghi K, Araghi T, Chutkan N, Araghi A. The Clinical Impact of Image Guidance and Robotics in Spinal Surgery: A Review of Safety, Accuracy, Efficiency, and Complication Reduction. Int J Spine Surg. 2021;15(s2):S10-S20.
  2. Urbanski W, Jurasz W, Wolanczyk M, Kulej M, Morasiewicz P, Dragan SL, et al. Increased Radiation but No Benefits in Pedicle Screw Accuracy With Navigation versus a Freehand Technique in Scoliosis Surgery. Clinical orthopaedics and related research. 2018;476(5):1020-7.
  3. O'Donnell C, Maertens A, Bompadre V, Wagner TA, Krengel W, 3rd. Comparative radiation exposure using standard fluoroscopy versus cone-beam computed tomography for posterior instrumented fusion in adolescent idiopathic scoliosis. Spine. 2014;39(14):E850-5.
  4. Dabaghi Richerand A, Christodoulou E, Li Y, Caird MS, Jong N, Farley FA. Comparison of Effective Dose of Radiation During Pedicle Screw Placement Using Intraoperative Computed Tomography Navigation Versus Fluoroscopy in Children With Spinal Deformities. Journal of pediatric orthopedics. 2016;36(5):530-3.
  5. Perdomo-Pantoja A, Ishida W, Zygourakis C, Holmes C, Iyer RR, Cottrill E, et al. Accuracy of Current Techniques for Placement of Pedicle Screws in the Spine: A Comprehensive Systematic Review and Meta-Analysis of 51,161 Screws. World Neurosurg. 2019;126:664-78 e3.
  6. Li HM, Zhang RJ, Shen CL. Accuracy of Pedicle Screw Placement and Clinical Outcomes of Robot-assisted Technique Versus Conventional Freehand Technique in Spine Surgery From Nine Randomized Controlled Trials: A Meta-analysis. Spine. 2020;45(2):E111-E9.
  7. Baldwin KD, Kadiyala M, Talwar D, Sankar WN, Flynn JJM, Anari JB. Does intraoperative CT navigation increase the accuracy of pedicle screw placement in pediatric spinal deformity surgery? A systematic review and meta-analysis. Spine Deform. 2022;10(1):19-29.
  8. Chan A, Parent E, Wong J, Narvacan K, San C, Lou E. Does image guidance decrease pedicle screw-related complications in surgical treatment of adolescent idiopathic scoliosis: a systematic review update and meta-analysis. Eur Spine J. 2020;29(4):694-716.
  9. Berlin C, Quante M, Thomsen B, Koeszegvary M, Platz U, Ivanits D, et al. Intraoperative radiation exposure to patients in idiopathic scoliosis surgery with freehand insertion technique of pedicle screws and comparison to navigation techniques. Eur Spine J. 2020;29(8):2036-45.

Reviewer 3 Report

In this article, the authors do nothing but describe the possible complications associated with surgery for spinal deformities in pediatric age. What innovations do the authors intend to bring to literature? Is there anything new compared to what was known in the past about this topic? What is the purpose for which this article should be published? Have the authors had direct experience of the complications mentioned? And in what percentage, if so?

Author Response

REVIEWER 3 – COMMENTS

Comment 6: In this article, the authors do nothing but describe the possible complications associated with surgery for spinal deformities in pediatric age. What innovations do the authors intend to bring to literature? Is there anything new compared to what was known in the past about this topic? What is the purpose for which this article should be published? Have the authors had direct experience of the complications mentioned? And in what percentage, if so?

Authors’ Response: Thank you for your comments. This review provides a comprehensive review of evidence regarding the management of complications associated with surgical management of paediatric spinal deformity surgery. This review incorporates recent published literature relating to bleeding, venous thromboembolism, pseudarthrosis, proximal junctional failure, pulmonary function pre-operatively and post-operatively, and regarding navigation. This review also reviews recently published practices and strategies for managing intraoperative neuromonitoring and intraoperative neurological events, for managing postoperative infection and superior mesenteric artery syndrome. This updated review of the evidence regarding the management of complications associated with surgical management of paediatric spinal deformity is therefore important.

We have added a sentence describing the reported rates (percentages) of revision surgery in a national spinal deformity service with reference, as follows:

The rate of revision surgery in a national spinal deformity service has been reported as 2.9% for removal or exchange of instrumentation, 1.5% for nonunion, 1.1% for infection, and 0.8% for adding-on deformity requiring extension of fusion (1).’

  1. Tsirikos AI, Roberts SB, Bhatti E. Incidence of spinal deformity surgery in a national health service from 2005 to 2018: an analysis of 2,205 children and adolescents. Bone Jt Open. 2020;1(3):19-28.

Round 2

Reviewer 3 Report

I appreciated the changes made to the original manuscript